# Clinical Significance of Combined Epithelial–Mesenchymal Transition Markers Expression and Role of Rac1 in Hepatocellular Carcinoma

**DOI:** 10.3390/ijms24021765

**Published:** 2023-01-16

**Authors:** Seung Kak Shin, Sujin Ryu, Seungyoon Nam, Seung Yeon Ha, Oh Sang Kwon, Yun Soo Kim, Se-Hee Kim, Ju Hyun Kim

**Affiliations:** 1Department of Internal Medicine, Gachon University Gil Medical Center, Gachon University College of Medicine, Incheon 21565, Republic of Korea; 2Department of Genome Medicine and Science, AI Convergence Center for Medical Science, Gachon Institute of Genome Medicine and Science, Gachon University Gil Medical Center, Gachon University College of Medicine, Incheon 21565, Republic of Korea; 3Department of Pathology, Gachon University Gil Medical Center, Gachon University College of Medicine, Incheon 21565, Republic of Korea; 4Gachon Medical Research Institute, Gachon University Gil Medical Center, Incheon 21565, Republic of Korea

**Keywords:** hepatocellular carcinoma, epithelial-mesenchymal transition, Rac1, Snail, p21-activated kinases 1

## Abstract

Epithelial–mesenchymal transition (EMT) has been implicated in cancer progression, invasion, and metastasis. We aimed to evaluate the correlations between clinicopathological characteristics and EMT markers in patients with hepatocellular carcinoma (HCC) who underwent surgical resection and to identify the key regulator in EMT process. Fresh-frozen HCC tissues and adjacent nontumor liver tissues from 30 patients who underwent surgical resection were provided by the Gachon University Gil Medical Center Bio Bank. Human HCC cell lines, Hep3B, SNU449, and Huh7 cells were transfected with Rac1 siRNA and exposed to hypoxic conditions. The combined EMT markers expression (down-expression of E-cadherin and overexpression of p21-activated kinases 1 (PAK1)/Snail) by Western blot in HCC tissues when compared to adjacent nontumor liver tissues was significantly associated with macrovascular invasion (*p* = 0.021), microvascular invasion (*p* = 0.001), large tumor size (*p* = 0.021), and advanced tumor stage (*p* = 0.015). Patients with combined EMT markers expression showed early recurrence and poor overall survival. In vitro studies showed that Rac1 knockdown decreased the expression of EMT markers including PAK1 and Snail in hypoxia-induced Hep3B cells and suppressed the migration and invasion of hypoxia-induced HCC cells. Rac1 may be a potential therapeutic target for inhibition of EMT process through the inhibition of PAK1 and Snail in HCC.

## 1. Introduction

Hepatocellular carcinoma (HCC) is the sixth most common cancer worldwide and the third leading cause of cancer mortality globally [1]. Surgical treatment such as resection or liver transplantation is an optimal treatment strategy offering potential cure and better long-term outcome in patients with HCC. Although the indications for surgical treatment have been expanded due to the improvement of treatment technology for HCC, recurrence and intrahepatic metastasis after surgery are still high [2].

Microvascular invasion (MVI) is known as a major risk factor for recurrence and intrahepatic metastasis after resection in patients with HCC [3]. However, the mechanism of vascular invasion in HCC and appropriate methods of suppressing it are still unclear.

Epithelial–mesenchymal transition (EMT) is a biologic process by which epithelial cells lose their cell polarity and cell-to-cell adhesion and convert into a mesenchymal phenotype, which has migratory and invasive properties [4]. EMT is essential for numerous developmental processes including mesoderm formation and neural tube formation as well as physiologic and pathologic processes, such as wound healing and the initiation of metastasis for cancer progression [5]. Several transcription factors such as Snail, Slug, and Zeb1 are related to EMT, and loss of E-cadherin expression and gain of N-cadherin expression are characteristic of EMT [6]. It is suggested that EMT results from multiple cancer signaling pathways including hypoxia, Wnt/β-catenin, transforming growth factor-β, and Hedgehog/Notch pathway, etc. [7]. Since EMT is believed to be a major mechanism by which cancer cells become migratory and invasive, a better understanding of the associations between EMT and HCC is crucial for the development of effective HCC therapies.

Rac1, a Rho GTPase family member, is known to regulate diverse cellular processes including cytoskeletal rearrangement, mitogenesis, membrane ruffling, and migration [8,9,10]. Rac1 is dysregulated in a variety of tumor types [11]. In addition, Rac1 promotes the EMT process of cancer stem-like cell phenotypes in gastric cancer, and thus, inhibition of Rac1 may prevent metastasis and augment chemotherapy for cancer [12].

P21-activated kinases (PAKs) are the major downstream effectors of small GTPases Rac and Cdc42, which are involved in the actin-based cytoskeletal remodeling [13,14]. Although Rac and PAK are important for physiological responses to growth factors in normal cells, it has been suggested that it is related to the EMT process in prostate cancer [15,16]. In addition, Rac1/PAK1 signaling contributes to high-glucose-induced podocyte EMT via promoting β-catenin and Snail transcriptional activities, which may be a potential mechanism involved in podocytes injury to stimuli under diabetic conditions [17].

The aims of this study were to evaluate the correlations between clinicopathological characteristics and EMT markers in patients with HCC who underwent surgical resection and to identify the key regulators in EMT process of HCC.

## 2. Results

### 2.1. Clinicopathologic Characteristics and EMT Markers of the Patients

The baseline characteristics of 30 patients are shown in Table 1. Mean age of patients was 58.0 ± 9.7 years, and 22 (77.2%) were men. Regarding the etiology of HCC, 80.0% of patients had hepatitis B virus infection, and 6.7% of patients had nonalcoholic steatohepatitis. Twenty-five (83.3%) patients had liver cirrhosis. Mean of tumor size was 4.0 ± 2.5 cm. Five (16.7%) patients had multiple tumors. Macrovascular invasion was observed in three (10.0%) patients. MVI was observed in nine (30.0%) patients. Seven (23.3%) and five (16.7%) patients were stage II and III, respectively. Means of alpha-fetoprotein and protein induced by vitamin K absence or antagonist-II were 1505.4 ± 3963.5 ng/dL and 3691.2 ± 14921.4 mAU/mL, respectively. Among patients, 73.3% down-expression of E-cadherin alone, 60.0% overexpression of PAK1 alone, 56.7% overexpression of Snail alone, and 30.0% combined EMT markers expression were observed, respectively.

### 2.2. Comparison of Clinicopathological Data between HCC Patients with Combined EMT Markers Expression and HCC Patients without Combined EMT Markers Expression

As shown in Table 2 and Figure 1, the combined EMT markers expression was significantly associated with large tumor size (*p* = 0.021), macrovascular invasion (*p* = 0.021), MVI (*p* = 0.001), and advanced tumor stage (*p* = 0.015). Overexpression of Rac1 (88.9% vs. 42.9%, *p* = 0.042) was frequently observed in patients with combined EMT markers expression compared with patients without combined EMT markers expression.

### 2.3. Sensitivity and Specificity of EMT Markers for Prediction of MVI in HCC

MVI is known to have a close relationship with postoperative recurrence after surgery, so it is important to predict MVI before surgery. We identified the related factors that are helpful in predicting MVI among EMT markers. The sensitivity, specificity, positive predictive value (PPV), and negative predictive value (NPV) of the combined EMT markers expression for prediction of MVI were 77.8%, 90.5%, 77.8%, and 90.5%, respectively (Table 3). The combined EMT markers expression was the most specific and good predictive performance for MVI. Among the single EMT markers, PAK1 or Snail was the sensitive method for predicting MVI, but its specificity was low.

### 2.4. Comparison of Recurrence-Free Survival (RFS) and Overall Survival between HCC Patients with Combined EMT Markers Expression and HCC Patients without Combined EMT Markers Expression

With a median follow-up of 25.5 months (1–82 months), 21 patients suffered from recurrence, and 9 of them relapsed within 1 year. Seven (77.8%) out of nine patients with combined EMT markers expression suffered from recurrence, and 71.4% (5/7) of the patients with combined EMT markers expression relapsed within 1 year. Since most of the patients with combined EMT markers expression relapsed within 1 year, the 1-year RFS rates of patients were lower in patients with combined EMT markers expression compared with patients without combined EMT markers expression (44.4% vs. 81.0%, *p* = 0.022; Figure 2B) although there is no significant difference in overall RFS (*p* = 0.352; Figure 2A). With a median follow up of 55.0 months (1–84 months), four patients died. Among four patients who died, three patients had positive combined EMT markers expression, and all of these three patients died within 1 year after surgery. The 1-year overall survival rates in patients with and without combined EMT markers expression were 63.5% and 94.1%, respectively. The log-rank test showed statistically significant differences between groups (*p* = 0.021; Figure 2C).

### 2.5. Rac1 Knockdown Decreases the Expressions of PAK1/Snail in Hypoxia-Exposed Hep3B Cells

Next, to investigate whether Rac1 regulates expression of PAK1 and other EMT-related proteins including Snail in hypoxia-exposed Hep3B cells, we performed Western blot analysis using Rac1 siRNA-transfected Hep3B cells in hypoxic conditions. As a result, hypoxia increased the protein levels of PAK1 and Rac1, whereas inhibition of Rac1 using siRNA downregulated expression of PAK1 in hypoxic conditions (Figure 3A). Furthermore, we examined the effect of Rac1 on the expression of other EMT-related proteins in hypoxia-induced Hep3B cells. Inhibition of Rac1 attenuated protein levels of Snail, N-cadherin, and Vimentin in hypoxia-induced Hep3B cells. Therefore, Rac1 may regulate hypoxia-induced EMT process via inhibition of PAK1 and EMT markers including Snail in HCC cells. Additionally, Rac1 knockdown repressed hypoxia-induced HIF-1α (Figure 3B).

### 2.6. Rac1 Knockdown Suppresses the Migration and Invasion of Hypoxia-Exposed HCC Cells

Given that inhibition of Rac1 repressed the expression of PAK1 and EMT-related markers including Snail, which is suggested to be related to metastasis and invasion in Hep3B cells, we investigated whether Rac1 modulates migration and invasion of HCC cells such as Hep3B, SNU499, and Huh7.

To examine whether Rac1 regulates the migration of HCC cells in hypoxic conditions, we confirmed efficient siRNA knockdown of Rac1 by Western blot in Hep3B, SNU499, and Huh7 cells (Figure 3A and Appendix A), and right after that, we performed the wound healing assay and transwell migration assay using Rac1 siRNA-transfected HCC cells. In the wound-healing assay, hypoxia induced the migratory ability of Hep3B, SNU449, and Huh7 cells by 60–119%, 56.8–122.9%, and 111–185.4%, respectively, whereas inhibition of Rac1 repressed hypoxia-induced migration of Hep3B, SNU449, and Huh7 cells by 70.7–78.1%, 86.5–91.1%, and 67.4–87.8%, respectively (Figure 4A). In the migration assay using transwells, hypoxia induced migration of Hep3B, SNU449, and Huh7 cells by 16.5–45.7%, 55.6–199%, and 11.1–82.4%, respectively, whereas Rac1 knockdown inhibited their migration by 43.3–58.2%, 58.2–76.1%, and 40.6–72.2% (Figure 4B). In addition, to investigate whether Rac1 controls the invasion of HCC cells, we utilized the invasion assay using transwells coated with Matrigel^®^. As shown in Figure 4C, hypoxia augmented the invasion abilities of Hep3B, SNU449, and Huh7 cells by 53.7–244%, 19.1–94.3%, and 10–120.6%, respectively, whereas Rac1 knockdown using siRNA attenuated their invasion by 62–86.1%, 63.2–88.5%, and 46.6–73.3%. Collectively, Rac1 knockdown using siRNA downregulated the migratory and invasive abilities of HCC cells induced by hypoxia, suggesting that inhibition of Rac1 may repress the migration and invasion of HCC cells through inhibiting the EMT process.

## 3. Discussion

Although the survival rate is improved due to the development of treatment for HCC, the recurrence rate remains still high after surgery in HCC patients with vascular invasion including MVI or advanced tumor stage. Therefore, it is necessary to understand the molecular mechanism in these patients and to apply appropriate preoperative prediction or specific treatment for them. In our study, combined EMT markers expression of down-expression of E-cadherin and overexpression of PAK1/Snail in HCC tissues when compared to adjacent nontumor liver tissues was related to vascular invasion including MVI and advanced tumor stage in HCC patients who underwent surgical resection. In addition, Rac1 knockdown decreased the expressions of PAK1- and EMT-related markers including Snail in hypoxia-exposed HCC cells, and the suppression of hypoxia-induced migration-invasive ability was confirmed by inhibition of PAK1 and Snail expression using Rac1 siRNA in HCC cells.

EMT is known as a complex process by which epithelial cells acquire the characteristics of invasive mesenchymal cells [18]. The EMT process is characterized by loss of epithelial markers such as E-cadherin, which is a key cell-to-cell adhesion molecule [19]. Snail is a zinc-finger binding transcription factor repressing cell adhesion proteins such as E-cadherin [20,21,22] and is known as a master regulator of the EMT implicated in key tumor biological processes including invasion and metastasis [23]. In addition, the recent study showed that PAK1 regulated the repressor activity of Snail by phosphorylating on Ser^246^ [24]. Previous studies suggested the relationship between clinicopathological characteristics and EMT-related markers in HCC patients who underwent surgical resection. Yang et al. demonstrated that low expression of E-cadherin alone was correlated with tumor aggressiveness, and the low expression of E-cadherin was correlated with high expression of Snail or Twist [25]. Zhai et al. suggested that the low expression of E-cadherin alone was related to the cancer stage and the local lymph node metastasis and reported on the relationship between low expression of E-cadherin and high expression of Vimentin [26]. Mima et al. reported that poor tumor differentiation, vascular invasion, extrahepatic recurrence, and shorter disease-free survival after surgery were confirmed in HCC patients with low expression of E-cadherin and high expression of Vimentin [27]. Miyoshi et al. reported that Snail overexpression alone in HCC tissue was associated with portal vein invasion and intrahepatic metastasis [28]. Ching et al. reported that PAK1 overexpression alone in HCC tissue was significantly associated with the presence of vascular invasion, poor tumor grade, advanced tumor stage, and shorter disease-free survival [29]. However, in our study, the down-expression of E-cadherin alone was confirmed about 73.3% in HCC patients who underwent surgical resection and was not significantly related to vascular invasion, tumor stage, or postoperative prognosis. Although overexpression of PAK1 alone and overexpression of Snail alone were sensitive methods to predict MVI in our study, but those were low specific for MVI and were not related to tumor stage and postoperative prognosis. The combined EMT markers expression, especially down-expression of E-cadherin and overexpression of PAK1/Snail was related to vascular invasion, tumor stage, and postoperative prognosis. Therefore, simultaneous identification of EMT markers and confirmation of their combined expression levels than single-marker analysis might be useful for understanding of patient’s condition more accurately and predicting the prognosis.

The EMT process of HCC can be identified more clearly in patients with macrovascular invasion. In our study, changes in combined EMT markers were confirmed in all three TNM stage IIIB patients with macrovascular invasion. However, our study confirmed that the EMT process is involved not only in macrovascular invasion but also in the MVI stage of HCC. MVI is defined as microscopic appearance of malignant cells or tumor emboli in the vascular cavities of endothelial cells or portal and hepatic venous systems [3]. MVI has been suggested as a risk factor of tumor recurrence after surgery in patients with HCC [30]. Previously, several studies showed the relationship between VEGF and MVI [31,32,33]. However, there is no significant relationship between VEGF expression and MVI in our study. The degradation of the basal membrane and herniation of the tumor cells to the capillary lumen were suggested as a key step of process in MVI [3]. In our study, EMT-related markers such as E-cadherin, Snail, and PAK1 were more relevant to MVI than VEGF, which is relevant to angiogenesis. In addition, the combined EMT markers expression was most specific and good predictive performance for MVI. Therefore, other treatment methods such as extended surgical resection or multidisciplinary management might be required in patients with combined EMT markers expression.

EMT is an important target for suppressing tumor metastasis and reducing drug resistance. There were several studies about the suppression of the EMT mechanism in HCC. MicroRNAs such as miR-34, miR-30A, miR-30B, and miR-153 are suggested as inhibitors of the EMT process through the suppression of Snail expression in HCC [34,35]. Reichl et al. confirmed that AXL receptor tyrosine kinase (AXL) was associated with the EMT process in HCC and suggested that EMT process could be suppressed by inhibiting AXL [36]. However, few drugs have been approved to effectively suppress the EMT process in HCC. Recently, Cao et al. confirmed that PAK1 and Snail were up-regulated in HCC cell lines, and PAK1 increased Snail expression and promoted EMT [37]. In addition, PAK1 is a key downstream effector of Small Rho GTPases Rac1 and CDC42 and which was related to cell morphogenesis, motility, mitosis, survival, and angiogenesis in various cancers [38,39]. Meanwhile, HCC is a hypermetabolic tumor, and rapid proliferation of HCC cells leads to an insufficient oxygen supply and subsequently generating a hypoxic microenvironment [40]. HIF-1α is known as a crucial regulator of EMT under hypoxic conditions [40], and previous study showed that the small GTPase Rac1 is activated in response to hypoxia and is required for the induction of HIF-1α protein expression [41]. Therefore, we examined the effect of Rac1 inhibition for suppression of hypoxia-induced EMT process through the inhibition of PAK1 and Snail. Our study demonstrated that the increase of Rac1, PAK1, and Snail expressions was observed in hypoxia-induced HCC cells, and Rac1 knockdown resulted in decrease of PAK1 and EMT-related markers including Snail, and suppressed the invasion and migration of HCC cells. We utilized Hep3B, Huh7, and SNU449 cells to investigate the regulatory effect of Rac1 on migration and invasion of various HCC cells. Hep3B and Huh7 cells are well-differentiated epithelial HCC cell lines, whereas SNU449 cells are poorly-differentiated mesenchymal HCC cells. In addition, Hep3B cells harbor non-sense p53 mutation, while Huh7 and SNU449 cells have p53 point mutation [42]. In our study, given that knockdown of Rac1 decreases migratory and invasive abilities of Hep3B, Huh7, and SNU449 cells, it seems that Rac1 can control migration and invasion of HCC cells regardless of the presence or absence of differentiation or the type of p53 mutation.

Inhibition of Rac1 may also be involved in a pathway that will prevent recurrence after surgical resection as a result of the inhibition of EMT process in HCC. Meanwhile, HIF-1α induced metabolic reprograming is suggested as one of the major molecular mechanisms that contributes to antiangiogenic agent and immune checkpoint inhibitor (ICI) resistance in HCC [43]. Our study also demonstrated that knockdown of Rac1 led to suppression of HIF-1α protein expression in hypoxia-induced HCC cells. These results suggest the possibility of a therapeutic target that can be utilized in overcoming the drug resistance of antiangiogenic agent and ICI.

The limitation of this study is that the analysis of this study was performed with a relatively small number of HCC samples. However, compared with previous studies, it is meaningful as a study that has simultaneously identified various EMT-related markers including PAK1 at the protein level and confirmed the clinical significance of combined EMT markers expression and the key regulator to inhibit EMT process.

In conclusion, simultaneous identification of EMT markers and confirmation of their combined expression levels in HCC tissues when compared to adjacent nontumor liver tissues are helpful in predicting MVI and prognosis after surgical resection in patients with HCC. Rac1 seems to be a key regulator for hypoxia-induced EMT process through inhibition of PAK1 and Snail in HCC. Therefore, Rac1 can be suggested as a potential therapeutic target for inhibiting EMT process, and new treatment strategy to inhibit postoperative recurrence in HCC patients with vascular invasion or advanced tumor stage.

## 4. Materials and Methods

### 4.1. Patients

The bio-specimen (fresh-frozen HCC tissues and their adjacent nontumor liver tissues) from 30 patients who underwent surgical resection between April 2012 and August 2015, and data used in this study were provided by Gachon University Gil Medical Center Bio Bank (Provided No. GBB2019-01), after approval from Institutional Review Board (GBIRB2018-449). Written informed consent was obtained from each participant. Samples were confirmed to be tumor or nontumor on the basis of histopathological assessment. Expression of EMT markers in these tissues was detected using Western blot. None of the patients had received treatment prior to surgical resection. HCC diagnosis was confirmed by histopathology according to American Association for the Study of Liver Diseases (AASLD) guideline [44]. The tumor grading was based on the criteria proposed by Edmondson and Steiner, and cancer stage was categorized according to eighth edition TNM-staging by the American Joint Committee on Cancer [45].

### 4.2. Follow-Up and Tumor Recurrence

After surgical resection, all patients were followed every 3 months to detect disease recurrence. Postoperative recurrence was defined as positive imaging findings in computed tomography or magnetic resonance imaging according to AASLD guideline compared with preoperative examinations or pathologic confirmation of recurrence by biopsy or resection [44].

### 4.3. Cell Lines and Hypoxic Condition

Human HCC cell lines, Hep3B, SNU449, and Huh7 cells were obtained from the Korean Cell line Bank (Seoul, Republic of Korea). Hep3B and SNU449 cells were grown in Roswell Park Memorial Institute-1640 medium (Welgene, Daegu, Republic of Korea) supplemented with 10% fetal bovine serum (FBS) (Welgene) and 1% penicillin streptomycin (PS) (Welgene). Huh7 cells were cultured in Dulbecco’s modified Eagle’s medium containing 10% FBS and 1% PS. For hypoxic incubation, cells were incubated in a hypoxic incubator (New Brunswick Scientific, Edison, NJ, USA) with a humidified environment consisting of 1% O_2_, 5% CO_2_, and 94% N_2_.

### 4.4. RNA Interference and Transfection

SMARTpool siRNA (a pool of four target specific siRNAs) against human Rac1 was obtained from Dharmacon RNAi Technologies (ThermoFisher Scientific, Socresby, Australia), and sequences for siRNA against human Rac1 are as follows: Rac1 #1 (5′-GUGAUUUCAUAGCGAGUUU-3′), #2 (5′-GUAGUUCUCAGAUGCGUAA-3′), #3 (5′-AUGAAAGUGUCACGGGUAA-3′), and #4 (5′-GAACUGCUAUUUCCUCUAA-3′). Cells were transfected with 50 nM SMARTpool siRNA or negative control siRNA (Dharmacon) using Lipofectamine 2000 (Invitrogen, Carlsbad, CA, USA) and incubated for 2–3 days until assays.

### 4.5. Western Blot Analysis

Tissues and cells were lysed in radioimmunoprecipitation assay (RIPA) buffer (50 mM Tris-HCl (pH 8.0), 150 mM NaCl, 0.5% sodium deoxycholate, 0.1% SDS, 1% NP-40) and centrifuged after sonication to collect the supernatant. Then, 20–30 μg proteins were separated in 8–12% SDS-PAGE gels and were transferred on PVDF membranes. After blocking in 5% non-fat milk for 1 h at room temperature, membranes were incubated with primary antibodies for overnight at 4 °C. Primary antibodies for Rac1 (Cell Biolabs Inc., San Diego, CA, USA), PAK1 (Cell signaling, Danvers, MA, USA), HIF-1α (Novus biologicals, Littleton, CO, USA), Snail (Cell signaling), Vimentin (Cell signaling), N-Cadherin (Cell signaling), VEGF (Santa Cruz Biotechnology, Santa Cruz, CA, USA), and glyceraldehyde 3-phosphate dehydrogenase (GAPDH) (Bethyl Laboratories, Inc., San Diego, CA, USA) were used. After washing three times with 0.1% Tween-20/PBS, membranes were incubated with horseradish peroxidase-conjugated secondary antibodies for 1 h at room temperature. After incubation in enhanced chemiluminescence reagent, proteins were detected by an Amersham Imager 600 (GE healthcare, Chicago, IL, USA). Protein expression profiles based on Western blot in HCC tissues were compared with that in paired adjacent nontumor tissues. Band intensity was quantified using ImageJ software (NIH, Bethesda, MD, USA). Fold change was displayed as the ratio of the target protein band to the GAPDH protein band.

### 4.6. Definition of Combined EMT Markers Expression

The combined EMT markers expression was defined as a simultaneous confirmation in down-expression of E-cadherin and overexpression of PAK1 and Snail in HCC tissues when compared to adjacent nontumor liver tissues by Western blot.

### 4.7. Wound HEALING assay

After transfection with siRNAs in 60 mm dish, cells were scratched using a pipette tip. Cells were incubated in normoxia or hypoxia for 24 h. Images were taken at 0 and 24 h with an Olympus CFX41 microscope (Hamburg, Germany). The calculation of wound area was analyzed using ImageJ software as described previously by Kim et al. [46]. The experiment was performed three times.

### 4.8. Transwell Migration Assay

siRNA-transfected cells were seeded in transwells (Corning Inc., Glendale, AZ, USA) with 0.2% gelatin-coated lower surfaces. Medium containing 20% FBS was used as attractant. After 6 h incubation in normoxia or hypoxia, cells were fixed and stained using crystal violet. Cells on the lower surface were mounted in mounting solution (Vectorshield, Vector Laboratories, Burlingame, CA, USA) and were counted with a light microscope (Olympus DP72, Hamburg, Germany). The experiment was performed three times.

### 4.9. Matrigel Invasion Assay

The lower surface and upper surface of a transwell were coated with 0.2% gelatin and 1 mg/mL Matrigel (BD bioscience, San Jose, CA, USA), respectively. siRNA-transfected cells were plated on transwells and were incubated in normoxia or hypoxia for 24 h. Medium containing 20% FBS was used as attractant. After fixing and staining, cells on the lower surface were counted with a microscope. The experiment was performed three times.

### 4.10. Statistical Analysis

The Mann–Whitney U test or Student *t*-test was used to compare the continuous variables between the two groups. Categorical variables were compared using Fisher’s exact test or χ^2^ test. The Kaplan–Meier method was used to compare the postoperative RFS and overall survival between HCC patients with combined EMT markers expression and HCC patients without combined EMT markers expression. The differences in the RFS and overall survival curves were compared using a log-rank test. All tests were two-tailed, and *p* < 0.05 was considered statistically significant. The analysis was performed using SPSS version 23.0 (IBM Corp., Armonk, NY, USA).

## Figures and Tables

**Figure 1 ijms-24-01765-f001:**
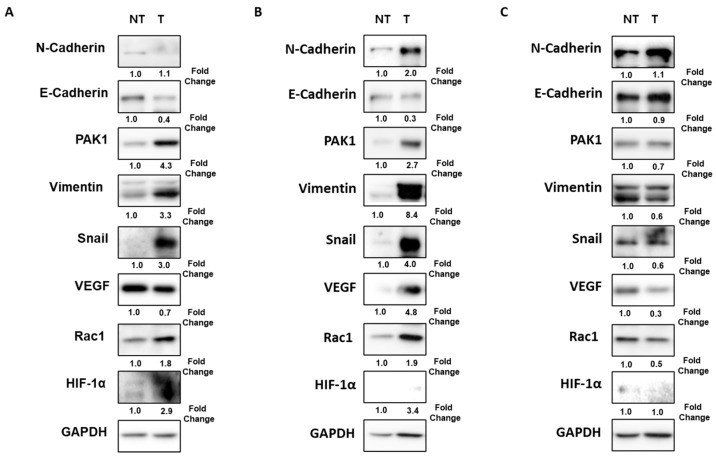
Combined EMT markers expression according to the tumor stage in patients with HCC. (**A**) Down-expression of E-cadherin and overexpression of PAK1/Snail in a 65-year-old man with 14 cm sized tumor and portal vein invasion (TNM stage IIIB). (**B**) Down-expression of E-cadherin and overexpression of PAK1/Snail in a 55-year-old man with 4 cm sized tumor and MVI (TNM stage II). (**C**) No combined EMT markers expression in a 71-year-old man with 3 cm sized tumor without vascular invasion (TNM stage IB). EMT, epithelial–mesenchymal transition; HCC, hepatocellular carcinoma; MVI, microvascular invasion; NT, non-tumor; T, tumor; PAK, p21-activated kinases; VEGF, vascular endothelial growth factor; HIF, hypoxia-inducible factor; GAPDH, glyceraldehyde 3-phosphate dehydrogenase.

**Figure 2 ijms-24-01765-f002:**
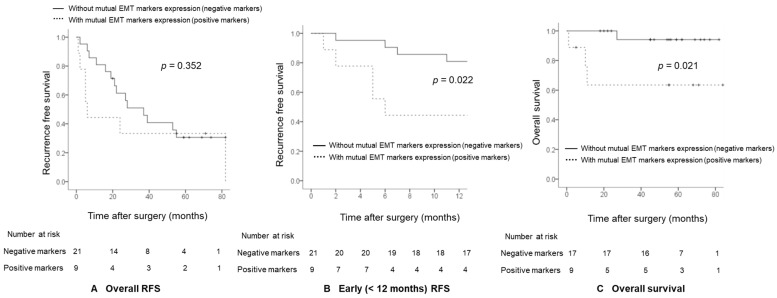
Kaplan–Meier curves for overall (**A**), early (<12 months) RFS (**B**), and overall survival (**C**) in HCC patients according to combined EMT markers expression. EMT, epithelial–mesenchymal transition; HCC, hepatocellular carcinoma; RFS, recurrence-free survival.

**Figure 3 ijms-24-01765-f003:**
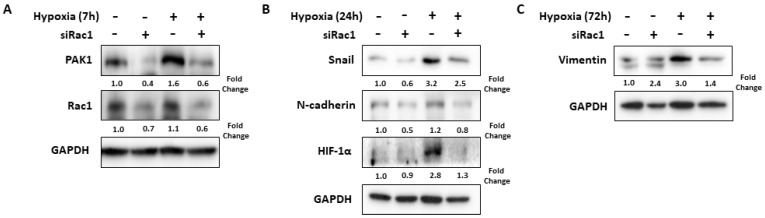
Rac1 knockdown decreases the expression of PAK1/Snail in hypoxia-exposed Hep3B cells. Hep3B cells were transfected with Rac1 siRNA or NC siRNA and incubated in normoxic or hypoxic conditions for 7–72 h. (**A**) Hypoxia induced increase of PAK1 and Rac1 expressions, and Rac1 knockdown decreased the expression of PAK1. (**B**) Hypoxia increased Snail, N-cadherin, and HIF-1α expression, and Rac1 knockdown decreased the expression of Snail, N-cadherin, and HIF-1α. (**C**) Hypoxia augmented Vimentin and Rac1 knockdown inhibited the expression of Vimentin. EMT, epithelial–mesenchymal transition; GAPDH, glyceraldehyde-3-phosphate dehydrogenase; HIF, hypoxia-inducible factor; NC, negative control; PAK, p21-activated kinase.

**Figure 4 ijms-24-01765-f004:**
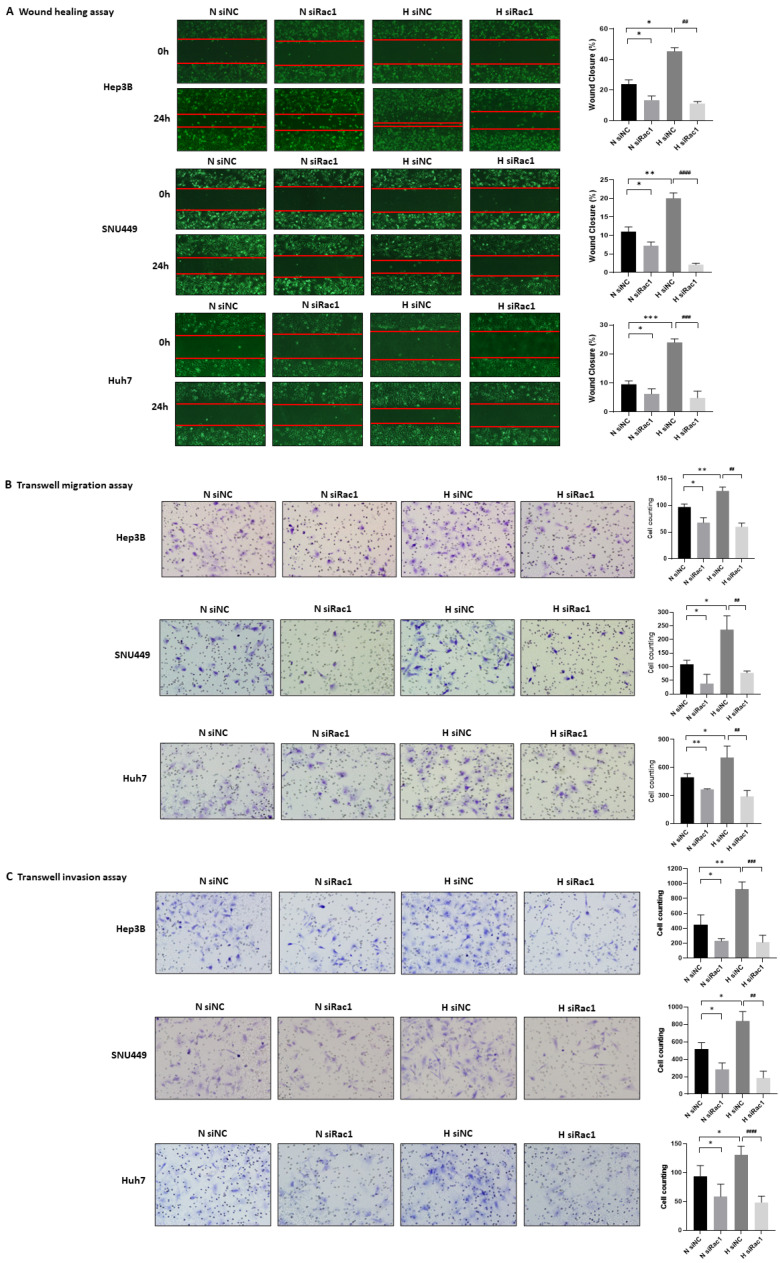
Rac1 knockdown suppresses the migration and invasion of hypoxia-exposed HCC cells. Hep3B, SNU449, and Huh7 cells were transfected with Rac1 siRNA and exposed to normoxic/hypoxic conditions for 6 (**B**)–24 h (**A**,**C**). (**A**) Wound-healing assay. Representative scratch-wound images showing the effect of Rac1 downregulation on the healing abilities of Hep3B, SNU449, and Huh7 cells (magnification: 40×) (left). Percentage of cells that migrated into the wound following Rac1 siRNA-transfected cells relative to NC siRNA transfected cells (right). (**B**) Transwell migration assay. Representative images showing the effect of Rac1 siRNA on migration of Hep3B, SNU449, and Huh7 cells through a Transwell chamber membrane (magnification: 200×) (left). Number of cells that migrated following Rac1 siRNA-transfected cells relative to NC siRNA transfected cells (right). (**C**) Transwell invasion assay. Representative images showing the effect of Rac1 siRNA on invasion of HCC cells through a Matrigel-coated Transwell chamber membrane (magnification: 200×) (left). Number of cells that invaded following Rac1 siRNA-transfected cells relative to NC siRNA transfected cells (right). N, normoxia; H, hypoxia; NC, negative control. * *p* < 0.05, ** *p* < 0.01, and *** *p* < 0.001 versus normoxia control group. *^##^ p* < 0.01, *^###^ p* < 0.001, and *^####^ p* < 0.0001 versus hypoxia-only group.

**Table 1 ijms-24-01765-t001:** Clinicopathologic characteristics and EMT markers of patients.

Variable	HCC Patients Who Underwent Surgical Resection (*n* = 30)
Age (years)	58.0 ± 9.7
Male gender	22 (77.3)
LC	25 (83.3)
Etiology	
HBV	24 (80.0)
HCV	1 (3.3)
HBV + HCV	1 (3.3)
Alcohol	1 (3.3)
NASH	2 (6.7)
Unknown	1 (3.3)
Tumor size (cm)	4.0 ± 2.5
Tumor number	
Single	25 (83.3)
Multiple	5 (16.7)
Tumor grade	
1	2 (6.7)
2	5 (16.7)
3	19 (63.3)
4	4 (13.3)
Macrovascular invasion	3 (10.0)
MVI	9 (30.0)
Glisson capsule invasion with perforation	4 (13.3)
Adjacent organ invasion	2 (6.7)
TNM stage	
IA	6 (20.0)
IB	12 (40.0)
II	7 (23.3)
IIIA	2 (6.7)
IIIB	3 (10.0)
IV	0 (0)
A-FP (ng/dL)	1505.4 ± 3963.5
PIVKA-II (mAU/mL)	3691.2 ± 14,921.4
Down-expression of E-cadherin	22 (73.3)
Overexpression of PAK1	18 (60.0)
Overexpression of Snail	17 (56.7)
Combined EMT markers expression	9 (30.0)

Values are expressed as number (%) or mean ± standard deviation; A-FP, alpha-fetoprotein; EMT, epithelial-mesenchymal transition; HBV, hepatitis B virus; HCC, hepatocellular carcinoma; HCV, hepatitis C virus; LC, liver cirrhosis; MVI, microvascular invasion; NASH, nonalcoholic steatohepatitis; PAK, p21-activated kinases; PIVKA-II, protein induced by vitamin K absence or antagonist-II.

**Table 2 ijms-24-01765-t002:** Comparison of clinicopathological data between HCC patients with combined EMT markers expression and HCC patients without combined EMT markers expression.

Variable	With Combined EMT Markers Expression (*n* = 9)	Without Combined EMT Markers Expression (*n* = 21)	*p*-Value
Age (years)	55 (50.0, 66.0)	59.5 (49.5, 67.5)	0.717
Male gender	7 (77.8)	15 (71.4)	0.999
LC	5 (55.6)	20 (95.2)	0.019
Etiology			0.897
HBV	9 (100)	15 (71.4)	
HCV	0 (0)	1 (4.8)	
HBV+HCV	0 (0)	1 (4.8)	
Alcohol	0 (0)	1 (4.8)	
NASH	0 (0)	2 (9.5)	
Unknown	0 (0)	1 (4.8)	
Tumor size (cm)	4.0 (3.2, 11.9)	2.7 (1.9, 4.0)	0.021
Tumor number			0.999
Single	8 (88.9)	17 (81.0)	
Multiple	1 (11.1)	4 (19.0)	
Tumor grade			0.713
1	0 (0)	2 (9.5)	
2	1 (11.1)	4 (19.0)	
3	6 (66.7)	13 (61.9)	
4	2 (22.2)	2 (9.5)	
Macrovascular invasion	3 (33.3)	0 (0)	0.021
MVI	7 (77.8)	2 (9.5)	0.001
Glisson capsule invasion with perforation	3 (33.3)	1 (4.8)	0.069
Adjacent organ invasion	2 (22.2)	0 (0)	0.083
TNM stage			0.015
IA	1 (11.1)	5 (23.8)	
IB	1 (11.1)	11 (52.4)	
II	3 (33.3)	4 (19.0)	
IIIA	1 (11.1)	1 (4.8)	
IIIB	3 (33.3)	0 (0)	
IV	0 (0)	0 (0)	
A-FP (ng/dL)	17.1 (5.0, 8134.0)	18.2 (6.8, 469.1)	0.665
PIVKA-II (mAU/mL)	1437 (16.3, 5169.5)	54.5 (22.0, 180.5)	0.600
Overexpression of Rac1	8 (88.9)	9 (42.9)	0.042

Values are expressed as number (%) or median (25th percentile, 75th percentile). A-FP, alpha-fetoprotein; EMT, epithelial–mesenchymal transition; HBV, hepatitis B virus; HCV, hepatitis C virus; HIF, hypoxia-inducible factor; LC, liver cirrhosis; MVI, microvascular invasion; NASH, nonalcoholic steatohepatitis; PAK, p21-activated kinases; PIVKA-II, protein induced by vitamin K absence or antagonist-II.

**Table 3 ijms-24-01765-t003:** Sensitivity and specificity of EMT markers for prediction of MVI in patients with HCC.

	Sensitivity (%)	Specificity (%)	PPV (%)	NPV (%)	AUROC (95% CI)	*p*-Value
Down-expression of E-cadherin	22.2	71.4	25.0	68.2	0.468 (0.242–0.694)	0.786
Overexpression of Snail	88.9	57.1	47.1	92.3	0.730 (0.542–0.918)	0.049
Overexpression of PAK1	100	57.1	50.0	100	0.786 (0.626–0.945)	0.015
Overexpression of VEGF	66.7	57.1	40.0	80.0	0.619 (0.399–0.840)	0.309
Down-expression of E-cadherin and overexpression of Vimentin	66.7	76.2	54.5	84.2	0.714 (0.503–0.925)	0.067
Combined EMT markers expression	77.8	90.5	77.8	90.5	0.841 (0.630–0.999)	0.004

AUROC, area under the receiver operating characteristic curve; CI, confidence interval; EMT, epithelial-mesenchymal transition; HCC, hepatocellular carcinoma; MVI, microvascular invasion; NPV, negative predictive value; PAK, p21-activated kinases; PPV, positive predictive value; VEGF, vascular endothelial growth factor.

## Data Availability

The data presented in this study are available on request from the corresponding author.

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
