# Peer review of "Clinical Significance of Combined Epithelial–Mesenchymal Transition Markers Expression and Role of Rac1 in Hepatocellular Carcinoma"

_ijms, 2023, doi:10.3390/ijms24021765_

Round 1

Reviewer 1 Report

In this manuscript, the authors investigated two epithelial-mesenchymal transition markers expression to predict Clinicopathological characteristics and explored Rac1 function in hepatocellular carcinoma. E-cadherin and Snail proteins have been heavily studied and used to predict outcomes, either use alone or combined, in various kinds of cancers including HCC. The authors analyzed the expression of proteins in 30 patients’ samples using western blot. The sample number is small. The sensitivity and Positive predictive value are a bit low.

Specific comments are listed below: 

-The overexpression and down expression of PAK1/Snail and E-cadherin were determined by comparing with adjacent nontumor tissues. But the GAPDH expression was not equal between the group. Did you quantify and normalized the bands and get the conclusion?

-The original images only show 18 patients. Can you provide the other 12?

- In Fig3. A, the knockdown of Rac1 without hypoxia is not very clear. Was this experiment repeated three times?

-In Fig3. B, the internal control GAPDH has a big difference in lanes 3 and 4. Though Snail protein looks down expressed, it could be the loading issue. Please use a new internal control or quantify the bands.

-In Fig3. C, what is the reason that the internal control switched to a-tubulin?

-Is there a western blot plot for Fig4 to confirm the knockdown of Rac1?

Reviewer 2 Report

You present an interesting work trying to understand the molecular basis of the epithelial-mesenchymal transition and its role in the metastasis process. An important point is the inclusion of patient samples, as you noted, the number is low but even so, from the results, some conclusions could be obtained. I would like to suggest a better description of your work, the characterization of the patient samples it is not easy to follow for those not involved in this type of research. The results presented in Table 3 are not easy to understand nor are the implications for the conclusion you're presented. Finally, you can discuss in more detail the implications of your results.

Reviewer 3 Report

In this study the authors attempt to assess the role of EMT in hepatocellular carcinoma, as well as the implication of Rac1 in the phenomenon.

1. The manuscript contains some grammar and syntax errors that need to be corrected (for examples, please refer to the uploaded file).

2. In the title and throughout the manuscript: I am not sure the term “mutual” reflects what the authors really mean. I believe "multiple" or "combined" would be more appropriate.

3. Page 2, ln95-96 “It is suggested that Rac1 pathway and hypoxia may be involved in the EMT mechanism of HCC”: It is not clear whether the statement presents published information or refers to findings of the current work. Please clarify and provide references, if necessary.

4. Please define abbreviations the first time they appear in the text and then use throughout.

5. siRNA against Rac1 does not seem to be efficient, since no reduction in Rac1 basal protein levels is shown for siRAc1-transfected cells in Figure 3A.

6. Why do the authors used 3 HCC lines? Are there differences in their molecular profile? This should be mentioned.

7. Most importantly, the limitations of the study are two: a. the small number of HCC samples and b. the lack of novelty in the molecular pathway investigated. The relationship between Rac1 and PAK 1 has been already established in several cell models.

Reviewer 4 Report

General Comments

Reviewed is the manuscript “Clinical significance of mutual epithelial-mesenchymal transition markers expression and role of Rac1 in hepatocellular carcinoma” submitted by Seung Kak Shin, et, al. The authors found that patients who had reciprocal EMT markers had early recurrence and a poor prognosis. Rac1 knockdown reduced the expression of EMT markers including PAK1 and Snail in hypoxia-induced Hep3B cells and inhibited the migration and invasion of hypoxia-induced HCC cells in in vitro research. The technique part is succinct and covers the key theories, which shows that the concept has been thoroughly understood. After minor modifications, the paper satisfies the requirements for publication since the authors have adequately explained their methodology and provided specifics about the performance increase in this study field.

Those comments are from a statistical perspective:

·         Increase the resolution or re-plot for Figure 2, the axis on them is hard to see.

·         Figure 2A was plotted but not cited in the manuscript.

·         According to the authors, patients with mutually expressed EMT markers had lower 1-year RFS rates than patients without mutually expressed EMT markers. How to explain why this difference is not significant in the total RFS?

·         Due to the number of tests that were conducted (E-cadherin, Snail, PAK1, VEGF), the multiple testing correction, such as FDR, is recommended.

Round 2

Reviewer 3 Report

This is the revised version of a previously submitted manuscript.

My concerns about the limitations of the study remain, but the authors have tried to address most of my suggestions.